# On the Role of Nanofluids in Thermal-Hydraulic Performance of Heat Exchangers—A Review

**DOI:** 10.3390/nano10040734

**Published:** 2020-04-11

**Authors:** Salah Almurtaji, Naser Ali, Joao A. Teixeira, Abdulmajid Addali

**Affiliations:** 1Transport and Manufacturing (SATM), School of Aerospace, Cranfield University, Cranfield MK43 0AL, UK; j.a.amaral.teixeira@cranfield.ac.uk (J.A.T.); a.addali@cranfield.ac.uk (A.A.); 2Kuwait Army, Kuwait Ministry of Defense, Safat 13128, Kuwait; 3Nanotechnology and Advanced Materials Program, Energy and Building Research Center, Kuwait Institute for Scientific Research, Safat 13109, Kuwait; nmali@kisr.edu.kw

**Keywords:** carbon nanotubes, colloidal, heat transfer, hybrid nanofluids, plate-fin, plate-frame, suspension

## Abstract

Heat exchangers are key components in many of the devices seen in our everyday life. They are employed in many applications such as land vehicles, power plants, marine gas turbines, oil refineries, air-conditioning, and domestic water heating. Their operating mechanism depends on providing a flow of thermal energy between two or more mediums of different temperatures. The thermo-economics considerations of such devices have set the need for developing this equipment further, which is very challenging when taking into account the complexity of the operational conditions and expansion limitation of the technology. For such reasons, this work provides a systematic review of the state-of-the-art heat exchanger technology and the progress towards using nanofluids for enhancing their thermal-hydraulic performance. Firstly, the general operational theory of heat exchangers is presented. Then, an in-depth focus on different types of heat exchangers, plate-frame and plate-fin heat exchangers, is presented. Moreover, an introduction to nanofluids developments, thermophysical properties, and their influence on the thermal-hydraulic performance of heat exchangers are also discussed. Thus, the primary purpose of this work is not only to describe the previously published literature, but also to emphasize the important role of nanofluids and how this category of advanced fluids can significantly increase the thermal efficiency of heat exchangers for possible future applications.

## 1. Introduction 

Depletable sources of fossil fuel such as crude oil, natural gas, and coal supply about 85% of the total global energy demands [1]. This percentage of energy consumption is expected to rise in the upcoming years, where according to the latest statistics on energy production and consumption, the worldwide total energy demand will increase by approximately 20% more by the year 2040 [2]. The majority of this energy is used to supply electricity and/or for the water desalination process. In the worldwide power generation market, gas turbines are becoming the preferred method for fulfilling most countries’ energy needs. This is because the technology itself has become mature enough and reliable for use, requires relatively short installation time, has a high cycle efficiency, relatively low operational and maintenance cost, and produces less CO_2_ and NO*_x_* emissions compared to other power generating systems [3,4]. Furthermore, the advances in Brayton cycle configurations have resulted in the widening of the range of possible applications, so today gas turbines can be seen employed in power plants, aviation, and marine propulsion. Gas turbines operate on the thermodynamic basis of the Brayton cycle, where it can be divided into two main categories, namely the open and the closed cycles. The major difference between the two is that in the open-cycle the working fluid (air) needs to be replaced with each complete cycle, whereas in the closed cycle, the heat transfer fluid (e.g., air or other gaseous fluid) is continuously reused. Thus, the closed cycle is seen as more cost-effective than the open system due to it requiring less fuel consumption while providing higher thermal transport efficiency [3,5]. Nevertheless, for propulsion applications (e.g., aircraft), open-cycle gas turbines are preferable because of their small construction scale, ease of load control, and higher turbine inlet temperature [6,7]. One of the key components that intensively influences the closed-cycle performance is the heat exchanger (HE), as it transports the required heat from a thermal source (e.g., solar, nuclear, or fossil) to the gas turbine cycle. In advanced closed-cycle system, reheaters, recuperators, and intercoolers, which are forms of HEs, are occasionally used to enhance thermal efficiency. Despite the achievements that started in 1935, when the closed-cycle gas turbine was first patented by Keller and Ackeret [5], researchers have come to a point where limited improvement in the cycle performance can be accomplished via modifying the design set-up, using different forms of heat exchanging devices, or adding turbulators to promote heat exchange [8,9]. Hence, they have concluded that to surpass the current limitations, an advanced type or category of heat transfer fluid needs to be employed [10]. This is when nanofluids, which were discovered in 1993 by Masuda et al. [11] and named by Choi and Eastman in 1995 [12], were seen as one of the primary solutions for overcoming the aforementioned cycle improvement limitations. The primary advantage of using nanofluids as working fluids, is that they exhibit significantly enhanced heat transfer characteristics when compared to their conventional counterparts [13,14]. 

A nanofluid is a suspension fabricated through homogenously dispersing nanoparticles (preferably <100 nm particle size and ≤1 vol. %) in a non-dissolving basefluid [15]. The particles themselves can be of pure metals, metallic oxides, carbides, carbon-based materials, alloys, or elemental compounds, whereas the hosting basefluids are usually made of any non-dissolving liquid such as water, ethanol, ethylene glycol (EG), oil, refrigerants, or a mixture made of two or more fluids [16]. Furthermore, the thermophysical properties (also referred to as the effective parameters) of the colloid greatly depends on the following points:
Nanoparticles:
Material type;Size and shape;Attraction/extraction characteristics with the hosting basefluid molecules;Volumetric concentration;Density;Specific heat capacity; andThermal conductivity.Basefluids:
Type;Temperature;pH value;Molecular attraction/extraction behaviour towards the dispersed particles;Density;Specific heat capacity;Viscosity; andThermal conductivity.Preparation route:
Single-step method; orTwo-step approach.Chemical or physical dispersion/s (if added).Nanofluid long- and short–term dispersion, kinetic, and chemical stabilities.

As such, it is crucial to carefully consider the previously mentioned points for each given heat transfer system to optimize its performance efficiency (i.e., power requirements and thermal transport). One of the common ways of doing this is by numerically analyzing the testing scenarios before actually conducting the experiments [17,18,19,20,21,22,23]. A measure of the relevance of the use of nanofluids as working media in HEs can be obtained from a survey of published works, according to the Elsevier’s abstract and citation database, Scopus, there are 2373 documents in total published between 1996 and 2020 [24]. Of these publications, 94.61% were published between the years 2009 and 2019 as illustrated in Figure 1. 

It is important to note that the data from Scopus were obtained by first screening the database for the words ‘Heat exchangers’, then the results were further refined to include the word ‘*Nanofluid*’ in the title, abstract, and/or keywords. Furthermore, investigating the reduction in the number of publications between the years 2011 to 2013, the authors have found that during the aforementioned period, researchers have focused essentially on improving the HE designs and the use of renewable sources for providing thermal energy to the system (e.g., solar or biomass). Additionally, the employment of nanofluids has extended beyond that of HEs to applications such as air purification systems [25], quenching media [26,27,28], liquid fuels enhancement [29], medical treatment [30], magnetic sealing [31], nano-lubricants [32,33], and many other usages. 

This paper will provide an overview of the operating principles of HEs together with a summary of the development of the design of HEs to incorporate the more efficient usage of nanofluids as working media. The main difference between the present review and those previously published is that this work focuses on two types of HEs, namely plate HEs (PHEs) and the plate-fin HEs (PFHEs). These HEs are well-known to have a wide range of applications in different industries. Furthermore, the paper focuses on nanosuspensions made of carbon nanotubes (CNTs), due to these particles having exceptional thermal properties when compared with other known nanomaterials [34,35]. For example, CNTs have extremely elevated thermal conductivity [36,37], large aspect ratio [38], lower density [39,40], lower erosion and corrosion surface effects [41], higher stability [37], and lower pressure drop and pumping power requirements in comparison to their counterparts [42,43]. Section 2 presents a review of HEs general operation theory together with an in-depth focus on two key types of heat transfer devices, such as the PHE and PFHE. In addition, Section 3 provides a background on nanofluids fundamentals and characteristics. Section 4 demonstrates the research work done on utilizing nanofluids in different types of HEs. Finally, Section 5 highlights the gap in present scientific knowledge that researchers will need to tackle in order to enable the commercial exploitation of these advanced types of fluids in heat exchange systems. 

## 2. Heat Exchangers 

A HE is a type of heat transfer device that is utilized for transferring the internal thermal energy between two or more fluids with different temperature levels [44]. In most cases, the HE design would not include any moving parts, thereby limiting weak points in the structure as well as frictional effects. However, in some exceptional cases, especially where compactness is a key requirement, the HE can include moving parts (e.g., rotatory regenerator and scraped surface HEs) [45,46]. As for the working fluid, it is usually in direct contact with the heat transfer surface, with conduction being the dominant heat transfer mode between the heat transfer surface and the fluid. Furthermore, there exist many different types of HEs, which are available for various industrial applications such as chemical reactors, refrigeration, power plants, food industry, petroleum, air-conditioning, heat recovery, and so forth [47,48]. The most familiar examples of HEs that are used in daily life are condensers, oil coolers, evaporators, and automotive radiators. There are many ways in which HEs can be categorised; for example industrial HEs can be classified based on the construction (i.e., tubular HEs, PHEs, extended surface HEs, and regenerative HEs), surface compactness, the phase of the working fluids (i.e., gas-liquid, liquid-liquid, and gas-gas), the arrangement of the fluid flow (i.e., parallel flow, counterflow, and crossflow), transfer processes (i.e., direct and indirect contact HEs), pass arrangements (i.e., single- or multi-pass HEs), and the heat transfer mechanisms (i.e., condensation and evaporation) [49]. Among all types of HEs, the plate and the plate-fin heat exchangers are the most widely used HEs throughout different applications. As such and due to the importance of these two types of HEs, the focus of the present review is specifically on these two types of HEs. 

Various techniques have been employed by researchers and designers to improve the thermal effectiveness of different HEs. Among all the important parameters that are associated with improving HEs performance, the working fluid is one of the most important aspects that caught the attention of researchers and designers [50]. This is due to the fact that utilizing a working fluid that possesses higher thermal conductivity would lead to enhancing the heat transfer rate. Thus, a number of researchers have investigated replacing conventional working fluids (i.e., water, ethylene glycol, oil, etc.) with advanced types of fluids that possess superior thermophysical properties such as nanofluids, which are suspensions of nano-sized particles (including oxide, metal-oxide, carbon nanotubes) in conventional working fluids [51,52,53,54,55]. This class of working fluids was mentioned for the first time, by Choi and Eastman [12] in 1995. Replacing the conventional working fluids with nanofluids would improve the thermal efficiency of existing HEs along with providing the designers with the capability of decreasing the overall weight and size of the device [56]. Moreover, the required pumping power and power consumption for transmitting the working fluid would also be subsequently reduced. 

In the following sub-sections, the PHEs and PFHEs are introduced, and their advantages and limitations are discussed in detail.

### 2.1. Plate Heat Exchangers

Historically, the first commercial PHE was introduced in 1923 by Dr. Richard Seligman [49], and this configuration remains extensively used in a range of devices in a variety of industries and applications. There are a number of different flow patterns and pass arrangements in PHEs, such as the series flow arrangement, single-pass looped arrangement, multi-pass with equal passes, and multi-pass with unequal pass. Figure 2 illustrates a schematic view of the different flow patterns and their pass arrangements. 

The main advantages gained from employing PHEs are: (1) they provide high turbulence, which results in added heat transfer coefficients and more fluid flow uniformity; (2) they eliminate the possibility of cross-contamination occurrence; (3) they provide true counterflow; and (4) they are easy to clean and maintain [49,56]. Nevertheless, there are some limitations associated with this type of HE. The main limitation would be that the PHEs must operate at a working condition of both low temperature and pressure (i.e., less than 149 °C and 300 psi). This is because the gaskets, which are designed for sealing, are typically made from elastomeric materials, and hence cannot withstand higher operational conditions than the ones specified previously. Moreover, the strength of the frame and the deformation resistance of the plate limit the permissible operating pressure. In addition to the aforementioned limitations, there is also the possibility of having leakages due to the use of gaskets; relatively low gasket lifetime as a result of the regular gasket removal for cleaning the plates; and the development of higher pressure drops in the system due to the narrow flow passages between the set of plates represent additional challenges to the design and operation of the PHEs.

Over the last decades, the markets have given rise to a strong demand for HEs that can overcome the pressure and temperature constraints, and thus scientists have continuously worked on maturing and developing the existing technology. This has led to introducing and manufacturing different types of PHEs. The main innovations and advantages of some of these PHE models are listed below:
-All-welded PHEs: The gaskets have been entirely eliminated, and as a result, the reliability of the HE has been enhanced by replacing a fully welded plate exchanger instead of the gaskets. This leads to eliminating the limitation associated with the operating pressure and temperature. The main constraint of this model is that no mechanical methods can be used for cleaning purposes, and therefore cleaning can only be achieved via chemical routes.-Wide-gap PHEs: Having the free-flow channel for highly viscous fluids and other products that contain coarse particles leads to eliminating the clogging problem that is usually encountered in HEs of the shell and tube type.-Free-flow PHEs: This unique design provides a wide flow path for fluids of high viscosity and fouling tendency and is also suitable for fluids that contain fibrous materials. This special design (free-flow design) can be considered as an improved design in comparison to the wide-gap PHEs. The main aspect of the design is that there is no contact point that can restrict the fluid flow in the flow path of the free-flow plates.

There are also other PHE models such as semi-welded or twin-plate HE, flow-flex tubular PHE, PHE with electrode plate, Supermax and Maxchanger PHE, plate-frame HE, and so forth. More detailed information about each model can be found in the *Heat Exchanger Design Handbook* [49]. 

While PHEs are particularly suitable for use in liquid-liquid applications in turbulent flow regimes, and they are widely used in different industries (i.e., heating and ventilating, food processing, breweries, oil and gas production, marine gas turbine, etc.), they are not recommended for use in gas-to-gas applications, vapor condensing under vacuum, and highly viscous fluids which pose flow distribution problems. 

### 2.2. Plate-Fin Heat Exchangers 

One type of compact HE is the PFHE. This category consists of parting sheets forming a stack of alternate flat plates with fin corrugations that are brazed together as a block. Figure 3 demonstrates a schematic view of the basic elements of a PFHE. Two main types of flow arrangement are available in PFHEs: crossflow and counterflow arrangement. These are schematically presented in Figure 4.

The compactness and lightweight of PFHEs are their most important features and it is for this reason that they have been widely employed in the aerospace industry. They are also used in cryogenics applications since they possess large heat transfer areas and are therefore designed to operate with small temperature differences. Other applications of PFHEs include petrochemical production, syngas production, automotive and locomotive, air separation, marine gas turbine, and oil and natural gas processing. 

One of the researchers in the field, Shah [57,58], discussed the most important features of PFHEs, and concluded the following: Fluid leakage possibilities do not exist or rarely occur, and as such there is no risk associated with the fluid mixing or contamination.PFHEs are designed for low-pressure applications (i.e., less than 1000 kPa).This class of HEs are mainly used for gas-to-gas applications and, in particular cases, in gas-to-liquid systems (e.g., the WR-21 marine propulsion gas turbine cycle).PFHEs offer high area density (i.e., up to approximately 6000 m^2^/m^3^).Various fins geometries (rectangular, tubular, offset strip, and wavy fin) can be utilized between the plates for various applications.PFHEs are designed for operating temperatures up to approximately 800 °C. The type of fin-to-plate bonding and the materials define the maximum operating temperatures.

PFHEs bring many advantages in practical applications. The main ones can be summarized as follows [59]: PFHEs provide superior thermal performance compared to their counterparts by using extended surfaces.PFHEs can operate effectively with temperature differences as low as 1 °C for single phase streams; while between multiphase streams, the temperature difference can be as low as 3 °C.For cryogenic applications, brazed aluminum PFHEs are the optimum choice due to the high surface compactness, the capability of handling multiple streams, and the highly desirable low-temperature properties at which they are able to operate.In cryogenic applications, a thermal effectiveness of the order of 95% or higher can be attained.PFHEs have large heat transfer surface per unit volume, and low weight per unit heat transfer.Exchanging heat between many process streams is possible in PFHEs.PFHEs can be used in different temperatures (from 0 to 800 °C) and pressures (up to 140 bar) by selecting the proper materials. However, they rarely get exposed simultaneously to a high temperature and pressure operating environment [58].

While PFHEs provide many advantages compared to other types of HEs, they also have some limitations. The most important limitations are the size and fouling build-up. The maximum size of a PFHE is limited by the dimensions of the brazing furnace and capacity of the furnace lifting. Moreover, the maximum size that can be brazed is also limited by the heating features of denser blocks. As for the fouling, since the passages in PFHEs are small, the applications of the PFHEs is limited to relatively clean streams since there is no easy way of cleaning PFHEs. Thus, the use of wavy fins is recommended in cases in which there is a high possibility of fouling [58]. It is important to note that the introduction of a fouled layer on the device surface will affect the fluid dynamics in the passages due to the modified surface wettability behavior, and with it the efficiency of the HE [60,61,62].

## 3. Nanofluids: Fundamentals and Characteristics 

The first theory to explain how the thermal properties of any conventional liquid could be enhanced through dispersing solid particles with higher thermal conductivity was the one provided by Maxwell in 1881 [63]. Maxwell’s concept was followed by experimental work performed on mm and µm scaled particles by Ahuja in 1975, Liu et al. in 1988, and by other researchers from Argonne National Laboratory (ANL) in 1992 [64,65,66,67,68]. Their results have shown significant improvement in the overall thermal conductivity of the fluids, but they have also noted that the particles within their as-prepared suspensions tended to largely agglomerate into each other, and hence clog small piping passages along the process. This is why, in 1993, Masuda et al. [11] explored the possibility of using ultra-fine particles to overcome the previous obstacle. The colloidal suspension that Masuda et al. [11] first introduced was later on called ‘*Nanofluids*’ by Choi and Eastman in 1995 [12]. Since nanoparticles have a large surface area exposed to the surrounding environment, these types of advanced suspensions have a far higher effective thermal conductivity compared to their mm and µm counterparts [12,69]. Examples of different types of nanoparticles and basefluids used in fabricating nanofluids are shown in Table 1. Although, now, scientists have advanced in the field of nanofluids to the point where hybrid nanofluids (i.e., nanofluids prepared with more than one nanomaterial) are being explored [70,71,72,73], optimizing the thermophysical properties and dispersion stability remains the main challenge with regard to the commercialization of such types of fabricated fluids.

### 3.1. Fabrication Approaches 

The method used for nanofluid preparation is the key step towards obtaining a homogenously dispersed suspension and hence can help towards providing a colloid of relatively ideal effective thermal conductivity and dispersion stability. There are two primary approaches for producing nanofluids, namely the single-step process (sometimes referred to as the one-step approach) and the two-step method [140,141]. The single-step route is a bottom-up approach, where the nanoparticles are synthesized and dispersed simultaneously in the basefluid through a single process [142,143]. The advantage associated with this fabrication approach is that it provides nanofluids of high dispersion stability and that the drying, storage, and transportation of nanoparticles are not needed for conducting the process. However, the drawback of the single-step method is that the formation of residuals due to uncompleted reactions is always present in the suspension. An example of such an approach is the vacuum evaporation onto a running oil substrate (VEROS) method, which was developed by Akoh et al. [144]. The previous process was afterwards modified by Wagener et al. [145], by using a magnetron sputtering device of high pressure to form dispersions of *Fe* and *Ag* particles. Eastman et al. [146] were also successful in developing the VEROS process (Figure 5) through condensing Cu vapor directly with a low-pressure vapor of EG to produce Cu—EG nanofluids. Other forms of the nanofluids one-step production method are the laser ablation, submerged arc nanoparticles synthesis system (SANSS), microwave irradiation, phase transfer, polyol method, physical vapor condensation, and plasma discharge. A detailed explanation of the aforementioned single-step approaches can be found in Mukherjee et al.’s [147]. 

On the other hand, the two-step method (Figure 6) uses physically or chemically pre-prepared nanoparticles, after which they are dispersed in a basefluid, with or without adding a surfactant, by one or more of the following processes:-Sonication [93,124,148,149,150].-Magnetic stirring [98,151,152,153].-Homogenizing [98,154,155].

As commercial nanoparticles are widely available, the two-step method is considered to be the most favorable and cost-effective approach for fabricating nanofluids, especially when a large quantity of the nanosuspension is desired [150,156]. Nevertheless, many researchers have experienced particle agglomeration and sedimentation within their nanofluids whilst employing the previously mentioned method [16]. Some have also reported variations in the thermophysical properties of their colloidal suspensions due to the difference in the temperature of the final product caused by the fabrication device and/or the surrounding environment [13,15,157]. 

### 3.2. Dispersion Stability 

As mentioned earlier, nanofluids have superior effective thermal conductivity compared to conventional working fluids. This notable enhancement in the thermal conductivity of the suspension is due to the high thermal conductivity of the dispersed nanoparticles within the basefluid of lower thermal conductivity. In the case where these particles become agglomerated and/or separated from the basefluid, through particle sedimentation, the nanofluid thermal conductivity degrades and hence loses its full potential. This is why the stability of the dispersion is one of the most important parameters which need to be optimized when fabricating such a class of fluids. Figure 7 illustrates the relation between the nanofluid stability and its effective thermal conductivity. 

There are three main aspects that influence the overall stability of the colloidal suspensions, namely the dispersion stability, chemical stability, and kinetic stability [147]. Dispersion stability describes the aggregation between the randomly distributed particles in the fluid, while chemical stability takes into account the chemical reaction that develops between the particles and their surrounding liquid environment. It is important to note that, under specific production conditions, the chemical stability of the colloidal suspension may be neglected (e.g., fabrication of nanofluids below the temperature point of a chemical reaction). Furthermore, the kinetic stability of the suspension is defined by the dynamic motion (also known as the Brownian motion) of the nanoparticles that are scattered within the basefluid. The fabrication approach impacts on the aforementioned three parameters (i.e., dispersion, chemical, and kinetic stabilities) and hence on the overall stability of the nanofluid; and is itself a function of the particles’ material, size, shape, and concentration; basefluid type, temperature, and acidity; and the type of surfactant used (if any). In order to determine the stability of nanofluids, researchers have worked on developing quantitative and qualitative techniques that have been adopted by many scientists in the field today [16,125]. Examples of these stability evaluation methods are: (1) light scattering technique, (2) sedimentation photographical method, (3) zeta potential analysis, (4) electron microscopy, and (5) 3-ω method. In addition, to enhance the stability of the colloidal, researchers have proposed the use of chemical routes, such as pH adjustment, particle surface modifications, and the addition of surfactant; and/or physical approaches such as ultrasonication, magnetic stirring, and homogenizer. Further details on stability improvement methods can be found in the works of Mukherjee et al. [147] and Ali et al. [16]. 

### 3.3. Nanofluids Thermophysical Properties 

The effective thermophysical properties of a nanofluid determine the rate of heat transfer that the fluid can provide to its hosting system [159]. These effective properties (e.g., effective thermal conductivity and effective viscosity) are influenced by the basefluid type and condition, particle characteristics, and volumetric concentration. When considering stability as a parameter of the thermophysical properties of the nanosuspension, researchers have found that such a factor does not affect the effective density nor the effective specific heat capacity of the colloidal suspensions [160,161,162,163]. This is due to the fact that the effective density of a nanofluid is constrained by the overall mass and volume of the mixture, while the effective specific heat capacity is directly governed by the concentration of the particles within the dispersion. It is important to note that the specific heat capacity of a nanofluid is usually lower than its basefluid [148,164]. The following equations can be used to accurately calculate both of the properties mentioned above, as reported in the literature [16]: (1)$$f_{V}=\frac{V_{\mathrm{np}}}{V_{\mathrm{np}}+V_{\mathrm{bf}}}\approx \frac{V_{\mathrm{np}}}{V_{\mathrm{bf}}}$$
(2)$$\rho_{nf}=f_{V}\times \rho_{\mathrm{np}}+\left(1-f_{V}\right)\times \rho_{\mathrm{bf}}$$
(3)$$C_{p_{nf}}=\frac{\rho_{bf}\times \left(1-f_{V}\right)}{\rho_{nf}}\times C_{p_{bf}}+\frac{\rho_{\mathrm{np}}\cdot f_{V}}{\rho_{\mathrm{nf}}}\times C_{p_{np}}$$
where $f_{V}$, $V_{\mathrm{np}}$, $V_{\mathrm{bf}}$, $\rho_{\mathrm{nf}}$, $\rho_{\mathrm{np}}$, $\rho_{\mathrm{bf}}$, $C_{p_{nf}}$, $C_{p_{bf}}$, and $C_{p_{np}}$ are the volumetric concentration, volume of nanoparticles, volume of the basefluid, nanofluid density, nanoparticles density, basefluid density, nanofluid specific heat capacity, basefluid specific heat capacity, and nanoparticles specific heat capacity, respectively. On the other hand, intensive research investigations on both effective thermal conductivity and effective viscosity of nanofluids have shown that these thermophysical properties are strongly dependent on the stability of the colloidal suspensions [89,165,166,167,168]. For example, in an unstable case, the value of the effective thermal conductivity of the dispersion degrades with settling time due to particle clustering and separation from the basefluid. As for the effective viscosity, an unstable suspension would result in the fluid being divided into multiple regions (i.e., two in the case of full particle separation, otherwise more) with the bottom section, due to gravitational force, being the most viscous of them all. Additional parameters that also contribute to the final effective viscosity and thermal conductivity of the suspension are the particle type, shape, and size; basefluid type, acidity, and temperature; addition of surfactant and/or functionalized materials; and the preparation approach used. Moreover, most of the proposed theoretical correlations that are available in the literature for nanofluids thermal conductivity and viscosity are limited by the experimental conditions from which they were derived (i.e., they are not universal equations) [16,56,169]. Readers can find full details on the historical and recent (up to the year 2019) developments in nanofluids’ effective thermal conductivity and effective viscosity correlations in the published work of Bakthavatchalam et al. [170]. For the aforementioned reasons, the most reliable determination of the two properties mentioned above can only be achieved accurately via experimental measurements, such as the following [133,171,172]: Effective thermal conductivity:
Cylindrical cell method;Steady-state parallel-plate method;Temperature oscillation approach;3-ω method;Thermal comparator method;Thermal constants analyzer approach;Flash lamp method; andTransient hot-wire method.Effective viscosity:
Capillary tube viscometer;Rotating viscometer;Capillary viscometer;Pressure differences over capillaries device; andTorsional oscillating cup.

It is important to note that, out of the previously mentioned devices, the transient hot-wire and the rotating viscometer methods are the two preferred approaches used for measuring nanofluids’ effective thermal conductivity and effective viscosity, respectively. In addition, it has been noted that the two aforementioned thermophysical properties always have higher values in the suspension form when compared to those of the basefluid used. 

Pozhar [173] investigated the viscosity of a nanofluid using a simplified version of the more rigorous Pozhar–Gubbins theory. His results supported the claim that the theoretical expression accurately modeled the properties of the nanofluid. The paper introduced the idea that the nanoparticles could penetrate the walls of the containing vessel and that this would enhance the layering effect at HE boundaries. The paper also presented results that showed that the layering of nanoparticles in the basefluid could increase its viscosity by a factor as high as 3 or even 4 at relatively low volume fractions. Xuan and Li [68] fabricated Cu–water and Cu–transformer oil nanofluids with the addition of laurate salt and oleic acid as surfactants, respectively. The stability of the samples was analyzed using a transmission electron microscope (TEM) device, while the effective thermal conductivity of the suspension was determined using a hot-wire apparatus. They found that adding 9 wt. % laurate salt and 22 wt. % of oleic acid caused the as-fabricated suspension of Cu–water to stabilize for 30 h and the Cu–transformer oil for 1 week, respectively. They further reported that increasing the volume fraction of particles in the suspension caused an increase in both effective thermal conductivity and effective viscosity. The authors linked this rise in thermophysical properties to the reduction in the distance between the scattered nanoparticles in the hosting basefluid. Khanafer et al. [174] developed a model to analyze the heat transfer performance of nanofluids inside an enclosure taking into account the solid particles’ dispersion. They solved the transport equations numerically using a finite-volume approach with an alternating direct implicit procedure. The results showed that the suspended nanoparticles increase the heat transfer rate at any given Grashof number, and the increase in heat transfer rate improved with the increase in nanoparticle volume fraction. However, there was a variance between different sets of results due to sedimentation and stability issues within their as-fabricated samples. Additionally, a series of experiments was carried out to measure the thermal conductivity and rheological properties of graphite-oil nanofluids by Wang et al. [175]. They used a scanning electron microscope (SEM) and a TEM device to observe and analyze the particles’ sizes and morphologies of their graphite-oil nanofluids. They also measured the thermal conductivity of their suspension through a transient hot-wire approach. Their findings indicated that parameters such as milling time, dispersant stability, graphite concentration, and the basefluid temperature had a significant effect on the enhancement of the thermal conductivity, with the particle volume fraction having the highest impact on the thermal property. For example, compared to the basefluid, their nanofluids showed an improvement in their thermal conductivities from 11 to 36%, when the solid volume fraction of graphite increased from 0.68 to 1.36%. 

## 4. Application of Nanofluids in Heat Exchangers

In Section 2, the applications and importance of HEs, especially plate and plate-fin HEs, have been presented and discussed in detail. It is known that HEs are used in many different industries for energy conversion, conservation, and saving. Thus, the efficiency of these HEs is of paramount importance. Various techniques have been so far employed by researchers to improve the heat transfer efficiency of HEs. Among these techniques, geometrical modification such as adopting fins to increase the heat transfer area, is one of the most widely used techniques in the literature. However, there are some drawbacks of using fins since they lead to an increase in the size and weight of the HEs [176]. Another method that can be employed to improve the heat transfer efficiency of HEs is to employ a new class of fluids which possess superior thermal properties compared to conventional working fluids. Amongst those new classes of working fluids, nanofluids are regarded as being very promising for the reasons described in Section 3 of this article. Various nanofluids containing oxide, metal oxide, CNT, and a combination of different nanoparticles (hybrid nanofluids) in different working fluids have been studied across the literature. 

It is well-known throughout the scientific community that CNT nanoparticles, including SW, DW, and MW, possess higher thermal properties compared to any other nanoparticles of metallic or metal oxide base. Figure 8 illustrates the thermal conductivity of some nanomaterials. As can be seen, CNT nanoparticles possess the highest thermal conductivity, and consequently the higher heat transfer rates. Therefore, many researchers have employed CNT-based nanofluids to augment the heat transfer efficiency of PHEs [177,178]. In the present review, the focus will be on the available literature that employed CNT-based nanofluids (mono and hybrid nanoparticles) to improve the heat transfer efficiency of plate HEs and plate-fin HEs.

### 4.1. Plate Heat Exchangers

#### 4.1.1. CNT-Based Nanofluids 

The effects of using a water-based nanofluid containing MWCNT nanoparticles on the heat transfer efficiency of a PHE in the application of milk pasteurization have been experimentally studied by Tabari and Heris [180]. They employed the two-step method for preparing the nanofluid samples in three different solid concentrations of 0.25, 0.35, and 0.55 wt. % by using a surfactant. They investigated the heat transfer coefficient (*h*) and the Nusselt (*Nu*) number. They performed the experiments over different ranges of Peclet number (350–1000). They found that increasing the solid concentration and Peclet (*Pe*) number leads to an augmentation of the *h* and *Nu*. They also reported that heat transfer performance is better at higher *Pe* numbers. They also presented the relative convective heat transfer coefficient (heat transfer of nanofluid to that of the basefluid) and reported that the maximum relative convective heat transfer coefficient (*h_nf_/h_w_*) took place at the *Pe* number of 1000 and solid concentration of 0.55 wt. % by 33%. Figure 9 presents the variations of the relative *h* with respect to the *Pe* number in different solid concentrations. 

Sarafraz and Hormozi [181] experimentally studied the effects of using MWCNT-water nanofluid as a working fluid on heat transfer and pressure drop of a PHE. They performed the experiments over a different range of *Re* numbers (700–25,000), solid concentrations (0.5–1.5 vol. %), and inlet fluid temperatures (50–70 °C). They found that increasing the Reynolds (*Re*) number and solid concentration results in the augmentation of the *Nu* number (Figure 10a). Moreover, they reported that increasing the fluid inlet temperature leads to a slight increase in the *Nu* number (Figure 10b). They also observed that increasing the *Re* number and solid concentration results in increasing the pressure drop. However, increasing the *Re* number leads to a more significant increase in the pressure drop compared to that of the solid concentration. They concluded that using MWCNT-water leads to having a better thermal performance of the PEH, while it imposes some penalty in terms of pressure drop and friction factor.

Goodarzi et al. [182] studied heat transfer and pressure drop in a counter flow corrugated PHE employing a MWCNT-water nanofluid in different solid concentrations (0.0 to 1.0 vol. %) and *Re* numbers (2500 to 10,000) under the turbulent flow regime. They investigated the effects of using nanofluids on the convective heat transfer coefficient, pressure loss and pumping power, and *Nu* number. They reported that the heat transfer performance increases as the *Re* number and solid concentration of nanoparticles increases. However, increasing the *Re* number and solid concentration results in increasing the friction factor, and as a result, increasing the pumping power. Finally, they reached the conclusion that replacing the MWCNT-water nanofluid with the basefluid (water) results in improving the heat transfer efficiency of the PHE. 

In a comparative experimental study, the effects of using pure water, MWCNT-water, and Al_2_O_3_-water nanofluid on the convective heat transfer and pressure drop in a PHE were investigated by Huang et al. [88]. They performed the experiments over a different range of *Re* numbers, solid concentrations, and flow velocities. They reported that, at a constant *Re* number, the nanofluids showed better heat transfer performance compared to that of the pure water. However, at a constant flow velocity, the heat transfer performance of the nanofluids deteriorated. They reported that the heat transfer enhancement of MWCNT-water nanofluid is higher than that of the Al_2_O_3_-water nanofluid. This would be because of the higher thermal conductivity of the MWCNT-water nanofluid. They also found that increasing the solid concentration of nanoparticles leads to decreasing the convective heat transfer coefficient, as can be seen in Figure 11. Moreover, they declared that increasing the solid concentration results in increasing the pressure drop and friction factor. Based on the experimental data, they also proposed a new correlation to predict the heat transfer and friction factor of the nanofluids. 

In another experimental study, performed by Kumar et al. [183], the heat transfer of a PHE using different nanofluids (TiO_2_, Al_2_O_3_, ZnO, CeO_2_, graphene nanoplate, and MWCNT-water) under the effects of variable spacing between the plates was studied. The spaces varied from 2.5 to 10 mm. The thermophysical properties of all the nanofluids investigated, including thermal conductivity, viscosity, density, and specific heat, were experimentally measured at different solid concentrations of 0.5, 0.75, 1.00, and 1.25 vol. %. As can be seen in Figure 12, all the measured thermophysical properties of the MWCNT-water nanofluid exhibited significantly higher values than those of the other nanofluids. Their results indicated that, at the spacing of 5 mm, the MWCNT-water nanofluid showed the highest heat transfer coefficient compared to the other nanofluids studied (see Figure 13a). The maximum heat transfer of the MWCNT-water nanofluid was 53% higher than that of the pure water. Moreover, the results showed that the minimum values of the pressure drop corresponded to the use of the MWCNT-water nanofluid (see Figure 13b).

#### 4.1.2. Hybrid Nanofluids Containing CNT Nanoparticles 

Another class of nanofluids, which received considerable attention from researchers, is hybrid nanofluids. Hybrid nanofluids contain a combination of different types of nanoparticles with a given ratio [184]. The main idea of using hybrid nanofluids is to enhance the thermophysical properties of the conventional working fluids. One of the widely used combinations of hybrid nanofluids involves the mixing of CNT nanoparticle with oxide and metal oxide nanoparticles. Many researchers investigated the thermophysical properties and heat transfer of hybrid nanofluids and reported that hybrid nanofluids showed better thermal performance compared to other types of nanofluids [32,33,71,73]. In this section, the available literature on the effects of using hybrid nanofluids on heat transfer performance of PHE will be reviewed and discussed. 

The effects of using MWCNT/Al_2_O_3_-water nanofluid, with the ratio of 1:2.5, on the heat transfer performance of a chevron corrugated PHE has been experimentally studied by Huang et al. [185]. They added a small amount of MWCNT to the nanofluid in order to increase the thermal conductivity of the mixture. They conducted the experiment with pure water, Al_2_O_3_-water, and the hybrid MWCNT/Al_2_O_3_-water nanofluid, and compared the results of the convective heat transfer coefficient and the pressure drop. They found that, at the same flow velocity, the thermal performance of the hybrid nanofluid was marginally superior to that of the pure water and Al_2_O_3_-water nanofluid (Figure 14a). They also noted that the pressure drop of the hybrid nanofluid is slightly lower than that of the Al_2_O_3_-water nanofluid (Figure 14b), whilst being higher than that of pure water. Based on the experimental data, they proposed a new correlation for predicting the *Nu* number. They concluded that the studied hybrid nanofluid would be a promising substitute for the water and Al_2_O_3_-water nanofluid in heat transfer applications. 

In another experimental study, Kumar et al. [186] investigated the heat transfer performance and exergy analysis of different hybrid nanofluids (Al_2_O_3_/MWCNT-water, TiO_2_/MWCNT-water, ZnO/MWCNT-water, and CeO_2_/MWCNT-water), with the ratio of 80:20, in different solid concentrations (0.25 to 2 vol. %) and at the temperature of 35 °C in a PHE. They began by measuring the thermophysical properties of the hybrid nanofluids, including thermal conductivity, dynamic viscosity, density, and specific heat. They reported that the maximum enhancement in thermal conductivity belonged to the CeO_2_/MWCNT-water hybrid nanofluid by 26.59%. Based on the Mouromtseff number [187], they found that the solid concentration of 0.75 vol. % is the optimum in which the maximum heat transfer would be achieved. Then, they investigated the effects of using the hybrid nanofluids on the heat transfer performance of a PHE and reported that, as expected, the CeO_2_/MWCNT-water nanofluid possessed the best performance compared to the other hybrid nanofluids studied. They also reported that the CeO_2_/MWCNT-water nanofluid showed the lowest exergy loss and total entropy generation. 

The effects of using a hybrid nanofluid containing Al_2_O_3_-MWCNT with different ratios (5:0, 4:1, 3:2, 2:3, 1:4, and 0:5), at the solid concentration of 0.01 vol. %, on the thermal performance of a counterflow corrugated PHE, have been experimentally studied by Bhattad et al. [188]. They performed the experiments over a range of the fluid inlet temperatures (10 to 25 °C) and flow rates (2.0 to 0.4 L/m), and studied the heat transfer coefficient, *Nu* number, pressure loss, and performance index. They found that the Al_2_O_3_-MWCNT-water hybrid nanofluid possessed a better thermal performance compared to Al_2_O_3_-water nanofluid. Moreover, variations of the *Nu* number versus *Re* number (Figure 15) revealed that the MWCNT-water nanofluid possessed the highest *Nu* number in all the *Re* numbers studied, while the Al_2_O_3_-water nanofluid showed the lowest *Nu* number. They also found than the maximum enhancement in heat transfer coefficient took place at the ratio of 0:5 (Al_2_O_3_:MWCNT) by 15.2%, while the minimum enhancement in heat transfer coefficient took place at the ratio of 5:0. They also reported that adding nanomaterials has a negligible effect on the pumping power and pressure loss.

In another experimental study, performed by Bhattad et al. [189], the effects of using various hybrid nanofluids (Al_2_O_3_-SiC, Al_2_O_3_-AIN, Al_2_O_3_-MgO, Al_2_O_3_-CuO, and Al_2_O_3_-MWCNT), with the ratio of 4:1, on the hydrothermal performance of a counterflow corrugated PEH have been investigated over a range of inlet fluid temperatures and flow rates. They reported that, amongst all the hybrid nanofluids studied, the Al_2_O_3_/MWCNT-water nanofluid possessed the highest heat transfer performance. The maximum enhancement was reported as 31.2%. However, using this hybrid nanofluid leads to a negligible increase in pumping power (0.08%). They concluded that the Al_2_O_3_/MWCNT-water hybrid nanofluid would be a good heat transfer fluid for enhancing the thermal performance of the PHE. 

#### 4.1.3. Other Types of Nanofluids 

In the previous sections, the effects of using CNT-based nanofluids and hybrid nanofluids containing CNT nanoparticles in PHEs have been reviewed and discussed in detail. Since the focus of the present review is on the CNT-based nanofluids, other types of nanofluids, containing oxide and metal oxide nanoparticles, will be only briefly reviewed, and a summary of the available literature will be presented in a table. 

Many studies have been performed on the effects of using various nanofluids, such as TiO_2_-water [190], ZnO-water [191], silver-water [192], graphene-water/EG [193], Al_2_O_3_-water [194], and so forth, on the thermal-hydraulic performance of different PHEs. The studies were performed on various solid concentrations, flow rates, and inlet fluid temperatures. Moreover, the effects of using nanofluids instead of conventional working fluids on the convective heat transfer coefficient, pressure drop and pumping power, *Nu* number, and the overall heat transfer coefficient have also been studied. The following conclusions have been reported by almost all the researchers; however, there are some differences between the reported values:It is reported by all the researchers that the *Nu* number and the convective heat transfer coefficient are enhanced by adding nanoparticles to the working fluids. Moreover, increasing the solid concentration of nanoparticles and the *Re* number leads to enhancing the *Nu* number and the convective heat transfer coefficient.Adding nanoparticles to the working fluids leads to increasing the dynamic viscosity of the resultant fluids (nanofluids) which, in turn, leads to increasing the pressure loss and pumping power. However, in some literature, it is reported that the increase in the pressure drop is negligible [190,195].Replacing the conventional working fluids with nanofluids will bring certain advantages from the heat transfer performance point of view. However, they impose some extra cost in terms of energy consumption; increasing the pressure drop leads to increasing the pumping power and energy consumption.

A detailed summary of the available literature on the effects of using different nanofluids on the heat transfer performance of various PHEs has been presented in Table 2. It is interesting to note that the widely used nanofluid in the literature is the Al_2_O_3_-water nanofluid. 

### 4.2. Plate-Fin Heat Exchangers

Unlike the plate HEs, the available literature on the effects of nanofluids on the thermal performance of plate-fin HEs is very limited. In the following, the effects of using different nanofluids on the thermal-hydraulic performance of plate-fin HEs will be reviewed and discussed. 

Aliabadi et al. [204] conducted an experimental and numerical investigation on the effects of vortex-generator and copper-water nanofluids on the heat transfer performance of a plate-fin HE. Figure 16 shows a schematic view of the studied plate-fin HE and the computational area. They studied the effects of *Re* number, solid concentration (0.1, 0.2, and 0.3 wt. %), and geometrical parameters on the heat transfer performance and pressure drop. They reported that increasing the solid concentration leads to increasing the heat transfer coefficient and pressure drop. However, geometrical parameters have a greater effect on heat transfer and pressure drop than the solid concentration.

In another experimental study by Aliabadi et al. [205], the thermal-hydraulic performance of copper-water nanofluids in a plate-fin HE was investigated. They studied seven different plate-fin channels as presented in Figure 17. They prepared the required nanofluids in different solid concentrations (ranging from 0.1 to 0.4 wt. %) using the single-step method. Then they measured the thermophysical properties of the prepared samples over different temperatures (298.51 to 313.15 K). The results revealed that increasing the volumetric flow rate and solid concentration leads to enhancing the heat transfer coefficient. It was found that, while the pressure drop in a lower solid concentration is approximately the same as that of the water, increasing the solid concentration results in experiencing some penalties in terms of an increase in pressure drop. It was also reported that the heat transfer enhancement and pressure drop increase was lower in the plain channel compared to other channels studied. Moreover, it was observed that decreasing the solid concentration resulted in an enhancement of the thermal-hydraulic performance of the plain channel. The authors noted that the best thermal-hydraulic performance, compared to the plain channel, was witnessed in a vortex-generator channel. They concluded that using nanofluids with lower flow rates as well as solid concentrations is more beneficial when compared to those with higher flow rates and concentrations. Figure 18 presents the variations of the thermal-hydraulic performance of the different channels studied versus solid concentration for different flow rates. 

The effects of various nanofluids (SiO_2_-, TiO_2_-, ZnO-, Fe_2_O_3_-, Al_2_O_3_-, and CuO-water) combined with a wavy channel on the thermal-hydraulic performance of a plate-fin HE have been studied by Aliabadi et al. [206]. The effects of using a parallel and corrugated wavy channel, different solid concentrations (0.1, 0.2, 0.3, and 0.4 wt. %), and the type of basefluid, including deionized water (DIW) and a mixture of DIW with EG in different mixing ratios of 100:0, 75:25, and 50:50, on the thermal-hydraulic performance have been experimentally studied. Two different definitions for the performance evaluation criteria have been employed for the thermal-hydraulic performance as follows: (4)$$\eta_{1}=\left(\frac{Nu_{i}}{Nu_{ref}}\right){\left(\frac{f_{i}}{f_{ref}}\right)}^{-1/3}$$
(5)$$\eta_{1}=\left(\frac{h_{nf}}{h_{bf}}\right){\left(\frac{\Delta P_{nf}}{\Delta P_{bf}}\right)}^{-1}$$
where the subscript *i*, *ref*, *bf*, and *nf* represent the used nanofluid, DIW, basefluid, and nanofluid, respectively. Moreover, *Nu*, *f*, *h*, and *ΔP* represent the Nusselt number, friction factor, convective heat transfer coefficient, and pressure drop, respectively. The results revealed that using the corrugated wavy channel leads to enhancing the heat transfer and pressure drop by 44.7% and 39.6%, respectively, compared to the parallel channel. Moreover, it was reported that the *Nu* number showed an increasing trend as the solid concentration decreases, while the friction factor decreases. Amongst the studied nanofluids, SiO_2_-DIW nanofluid showed the best heat transfer performance. Moreover, it was found that using a mixture of DIW and EG with SiO_2_ nanoparticle showed better thermal-hydraulic performance compared to the SiO_2_-DIW nanofluid. The ratio of 75% DIW and 25% EG was reported as the optimum ratio for having the best thermal-hydraulic performance (Figure 19). 

In another experimental study, the thermal-hydraulic performance of Cu-water nanofluid in a range of solid concentrations (0.1 to 0.4 wt. %) in a plain channel and vortex-generator channel (Figure 20) in a plate-fin HE were investigated by Aliabadi [207]. Based on the results, increasing the solid concentration leads to increasing the thermal-hydraulic performance of the plain channel plate-fin HE (Figure 21a). It was also revealed that applying a vortex-generator leads to having a higher thermal-hydraulic performance of the plate-fin HE compared to the nanofluid. Moreover, it was revealed that applying vortex-generator and nanofluid leads to having the highest thermal-hydraulic performance of the HE (Figure 21b). 

There are a number of other similar studies that investigated the effects of using different passive techniques, including using nanofluids, vortex-generators, winglets, and perforations, on the thermal-hydraulic performance of plate-fin HEs [208,209,210,211]. These authors performed studies over a range of nanofluids and different geometries of vortex-generators. Their results revealed that applying nanofluids as a heat transfer fluid results in having a better thermal-hydraulic performance of the HEs. Moreover, combining the nanofluids with other passive techniques leads to having higher thermal-hydraulic performance. Table 3 presents a summary of the published literature on the effects of using different passive techniques on the thermal-hydraulic performance of plate-fin HEs.

## 5. Discussion and Future Direction

This survey shows that one of the effective ways to enhance the thermal-hydraulic performance of HEs, especially plate and plate-fin HEs, is to replace the conventional working fluids with nanofluids. Amongst the different types of nanofluids, CNT-based nanofluids attracted the most attention due to their unique features such as enhanced thermal conductivity and specific heat, a higher heat transfer rate, low pumping power and pressure drop, and higher stability when compared to other nanofluids. In addition, they caused the least erosion, corrosion, and collaging [213], all of which are highly critical parameters in the operation of compact HEs. 

Available literature revealed that adding nanoparticles to the conventional working fluids leads to enhancing the thermophysical properties and thermal performance of the basefluid. As a result, using nanofluids as a working fluid in HEs leads to an improvement in the thermal-hydraulic performance of the HEs. Moreover, increasing the solid volume fraction of the nanoparticles, at a constant Re number, leads to enhancing the heat transfer rate. The simultaneous increase in the *Re* number and solid volume fraction results in increasing the friction coefficient, which in turn leads to an increase in the pumping power and pressure loss. 

A small number of papers reported that there is an optimum solid concentration at which the thermal-hydraulic performance reaches a maximum level. Increasing the solid concentration any further results in a deterioration of the thermal-hydraulic performance. However, more studies need to be conducted on a wide range of nanofluids and for a variety of engineering applications in order to discover the optimum parameter combinations. Furthermore, the effects of the sedimentation of, and corrosion by, nanofluids on the long-term operation of HEs also needs to be explored, as it is part of the feasibility chain for commercializing such types of fluids. Some authors have reported fouling formation due to particle deposition on the surface. Such a thin layer can cause changes to the liquid-surface wettability behavior of the HE, and hence can have a negative influence on the flow dynamics of the suspension. In addition, the relation between the enhancement in heat transfer performance of HEs and their geometric parameters, such as the fin and/or plate thickness, plate pitch, corrugation angle, and corrugation pattern, needs to be investigated given the wide range of available designs and sizes of these devices on the market. The aforementioned points are likely to be the main direction of future research into the applications of using nanofluids, especially CNT-based nanofluids, in HEs. 

## 6. Concluding Remarks

This paper presents a review of the available literature on the effects of using nanofluids, especially CNT-based nanofluids, on the thermal-hydraulic performance of two special types of compact HEs; plate and plate-fin HEs. The fundamentals of HEs have been presented and discussed, as have the effects of using different nanofluids as well as other passive techniques to improve the thermal-hydraulic performance of plate and plate-fin HEs. Based on the available literature, employing nanofluids instead of conventional working fluids results in enhancing the heat transfer performance of HEs. However, it also leads to a higher system pressure drop. The survey shows that using CNT-based nanofluids results in a higher heat transfer enhancement together with a smaller increase in the pressure drop in comparison with conventional fluids. Furthermore, combining other passive techniques, such as using vortex-generators, winglets, and perforation, with using nanofluids results in having a better thermal performance of the HE. However, the fitting of these turbulence promoting features leads to higher pressure drops in the system compared to using the nanofluids in isolation. In conclusion and based on the available literature, the usage of CNT-based nanofluids is highly effective in improving the heat transfer performance of HEs, especially plate and plate-fin HEs, since they cause the highest heat transfer enhancement at the lowest pressure drop increase.

## Figures and Tables

**Figure 1 nanomaterials-10-00734-f001:**
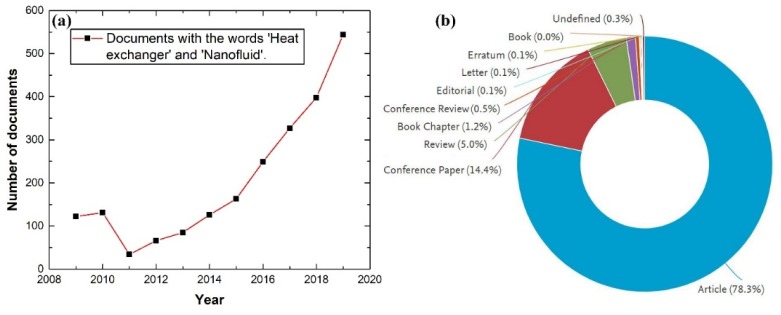
Data obtained from the Scopus database for the words ‘Heat exchanger’ and ‘Nanofluid’ from 2009 to 2019, where (**a**) shows the number of publications per year, and (**b**) demonstrates the type and percentage of these publications.

**Figure 2 nanomaterials-10-00734-f002:**
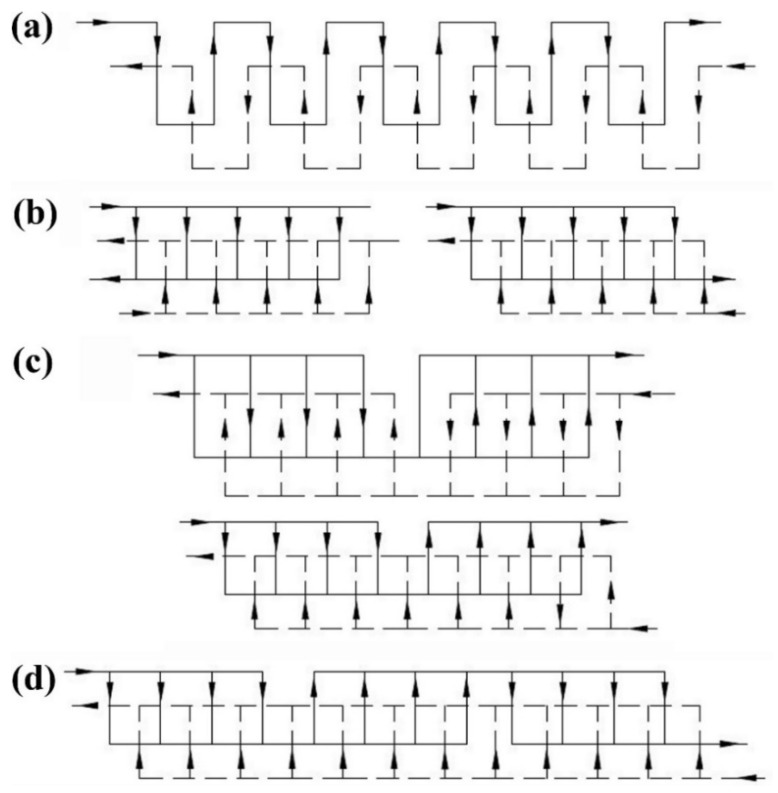
A schematic view of different flow patterns and pass arrangements in plate heat exchangers (PHEs), where (**a**) series flow, (**b**) single-pass looped, (**c**) multi-pass with equal pass, and (**d**) multi-pass with unequal pass [49].

**Figure 3 nanomaterials-10-00734-f003:**
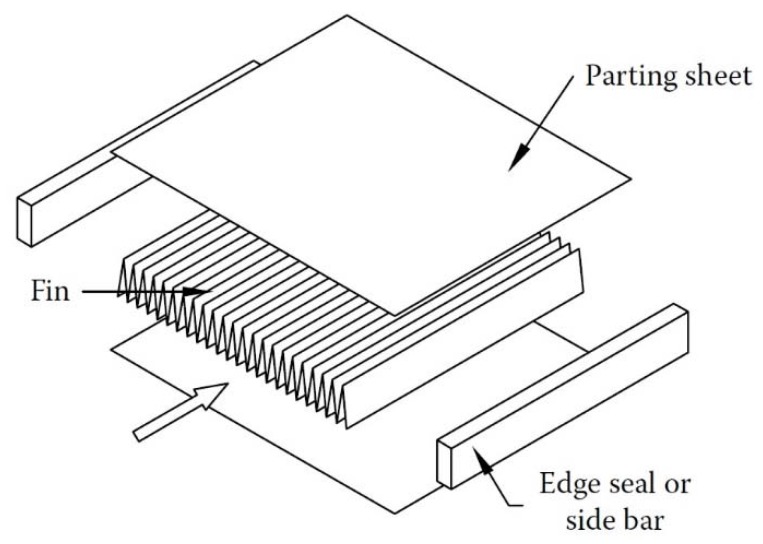
A schematic view of different elements of a plate-fin heat exchanger (PFHE) [49].

**Figure 4 nanomaterials-10-00734-f004:**
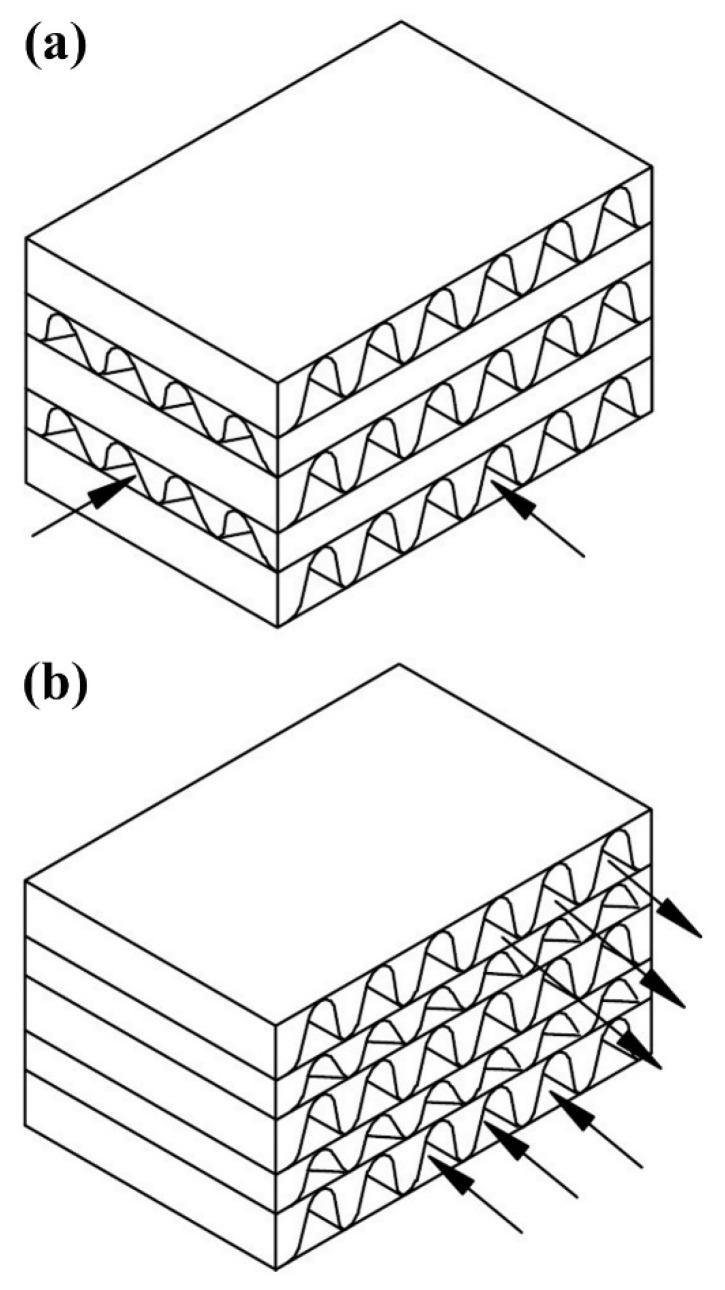
A schematic view of the two types of flow arrangement in a PFHE, where (**a**) Crossflow, and (**b**) Counterflow arrangement [49].

**Figure 5 nanomaterials-10-00734-f005:**
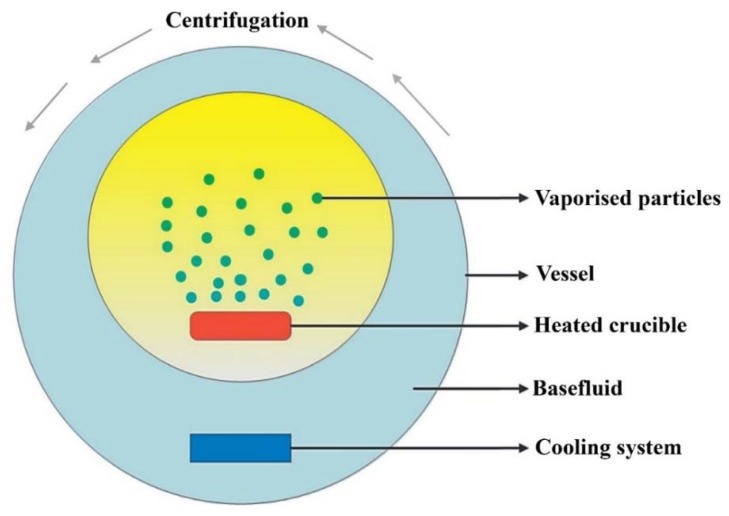
Eastman et al. one-step method for nanofluids fabrication. Reproduced with permission from [146]. Cambridge University Press, 2011.

**Figure 6 nanomaterials-10-00734-f006:**
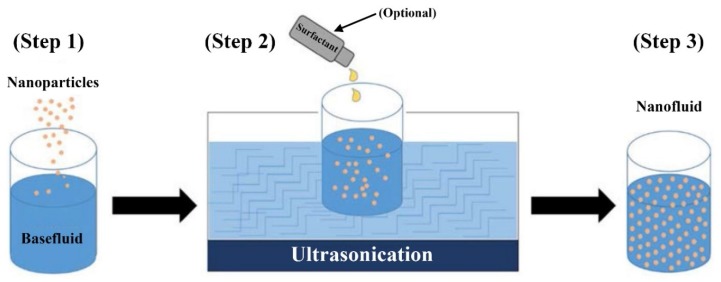
Illustration of the two-step nanofluid fabrication method using an ultrasonicator device [158].

**Figure 7 nanomaterials-10-00734-f007:**
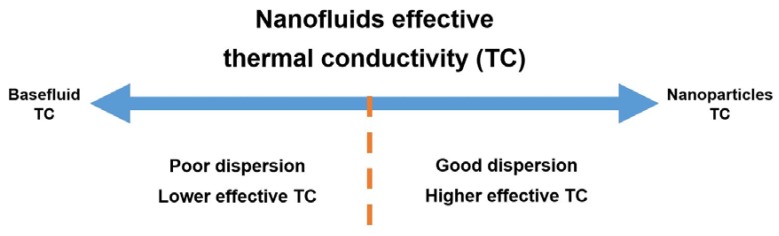
Relation between nanofluid stability and its effective thermal conductivity.

**Figure 8 nanomaterials-10-00734-f008:**
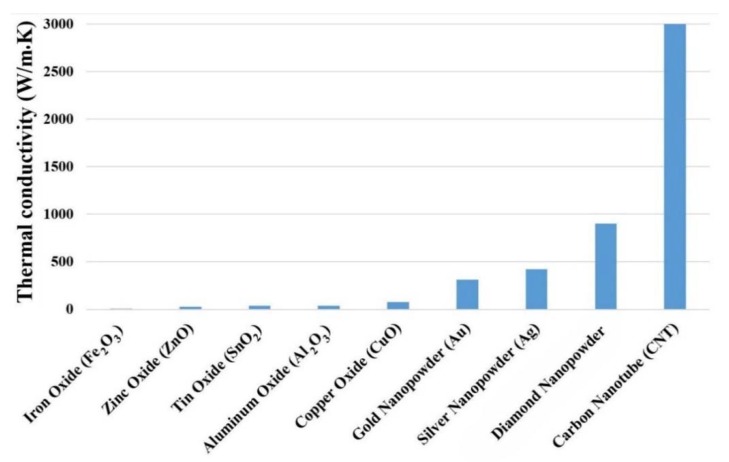
Thermal conductivity of selected nanomaterials [179].

**Figure 9 nanomaterials-10-00734-f009:**
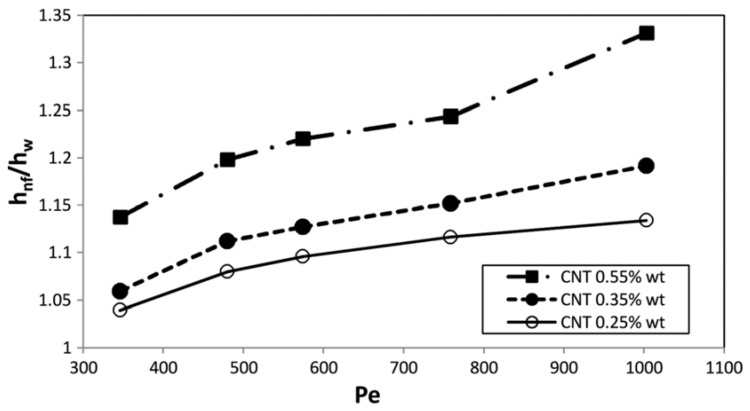
Variations of relative convective heat transfer coefficient versus Peclet number in different solid concentrations. Reproduced with permission from [180]. Taylor & Francis, 2015.

**Figure 10 nanomaterials-10-00734-f010:**
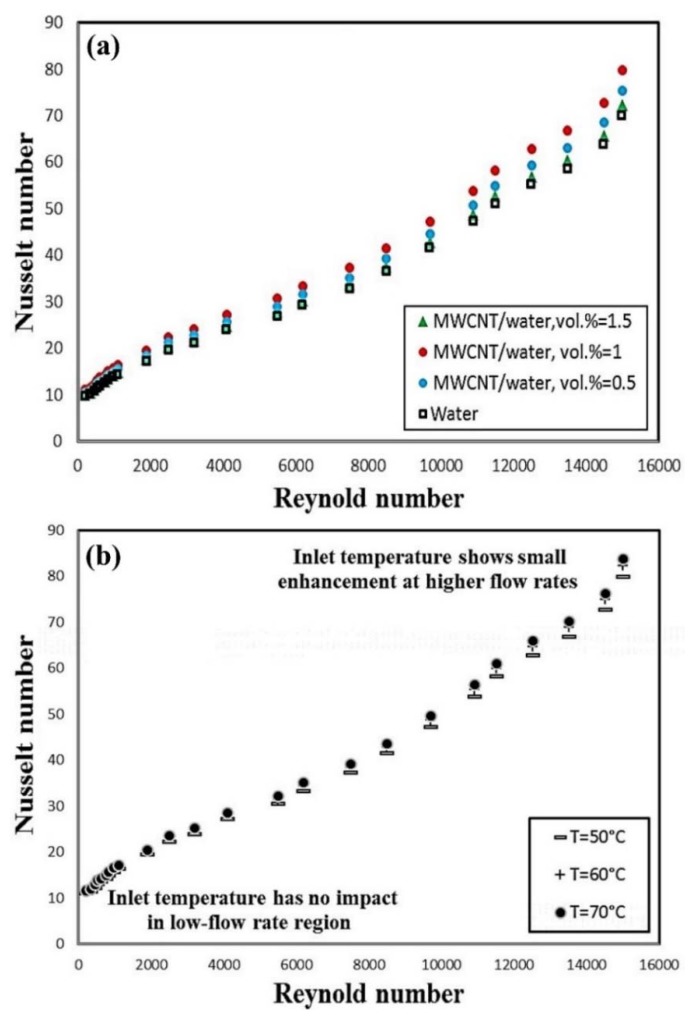
(**a**) Variations of *Nu* number of the nanofluid and water versus *Re* number, and (**b**) Variations of the *Nu* number versus *Re* number in different inlet fluid temperature. Reproduced with permission from [181]. Elsevier, 2016.

**Figure 11 nanomaterials-10-00734-f011:**
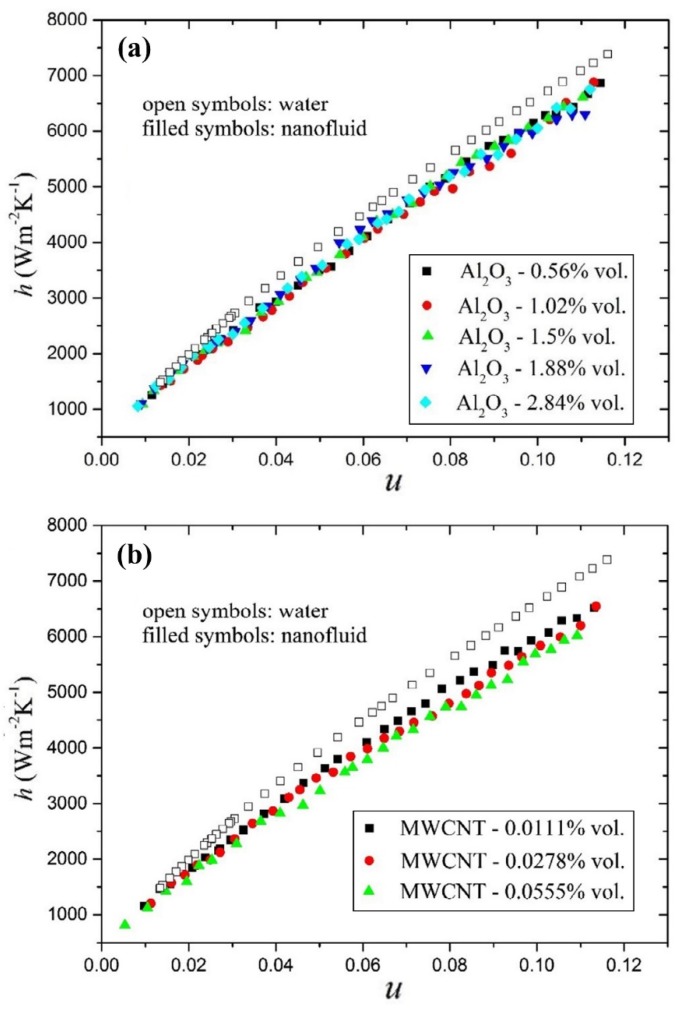
Variations of the convective heat transfer coefficient versus flow velocity (*u*) in different solid concentrations for: (**a**) Al_2_O_3_-water nanofluid, and (**b**) MWCNT-water nanofluid. Reproduced with permission from [88]. Elsevier, 2015.

**Figure 12 nanomaterials-10-00734-f012:**
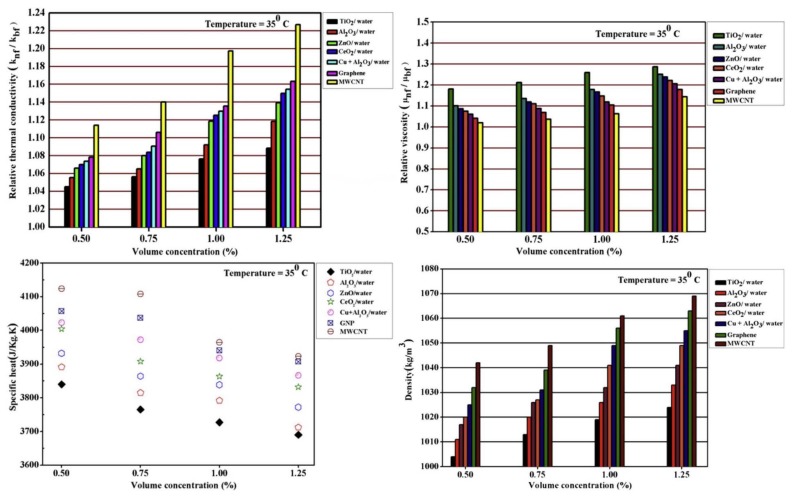
Variation of the relative thermophysical properties of the studied nanofluids. Reproduced with permission from [183]. Elsevier, 2016.

**Figure 13 nanomaterials-10-00734-f013:**
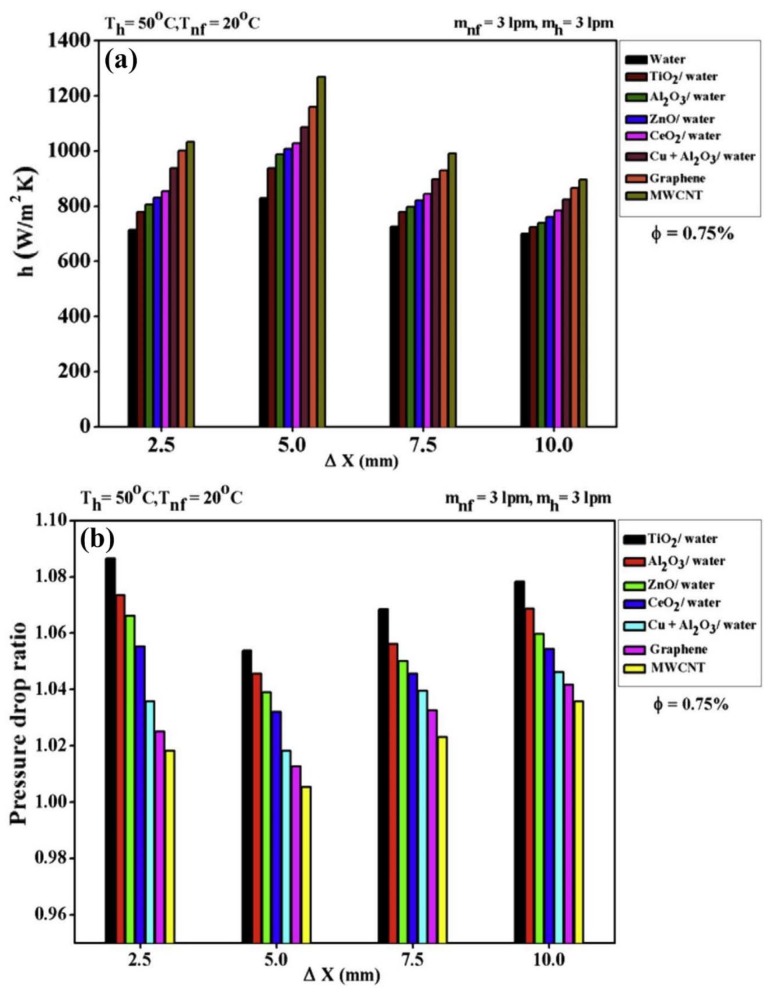
Variations of (**a**) the convective heat transfer coefficient, and (**b**) pressure drop versus the spacing value at the solid concentration of 0.75 vol. %. Reproduced with permission from [183]. Elsevier, 2016.

**Figure 14 nanomaterials-10-00734-f014:**
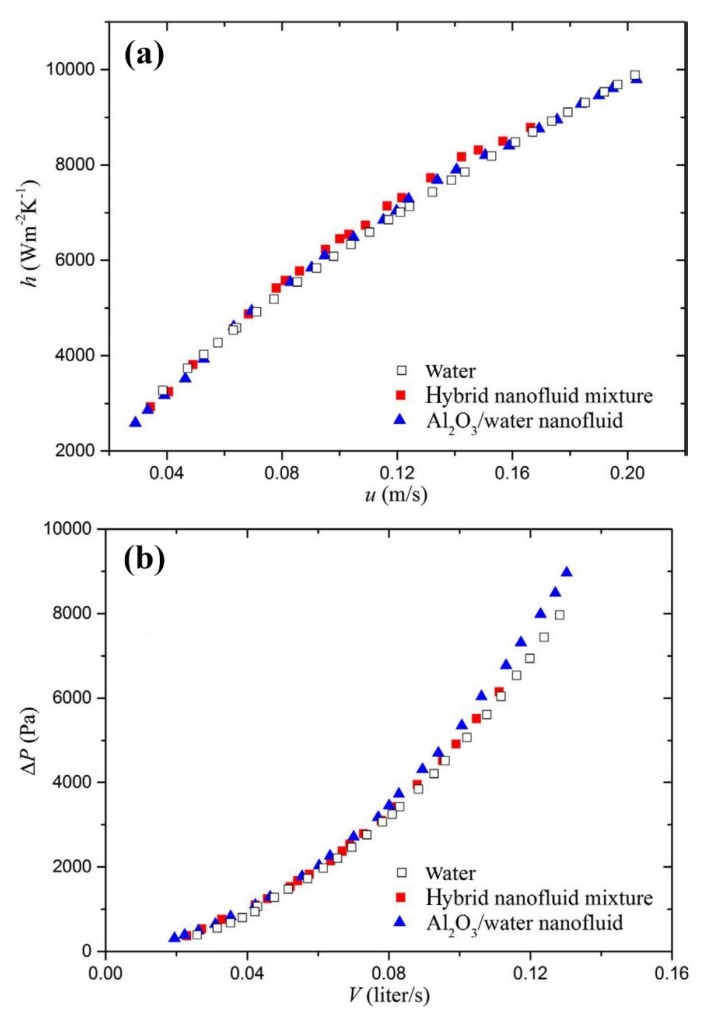
Variation of the (**a**) convective heat transfer coefficient, and (**b**) pressure drop versus fluid velocity for the three different fluids. Reproduced with permission from [185]. Elsevier, 2016.

**Figure 15 nanomaterials-10-00734-f015:**
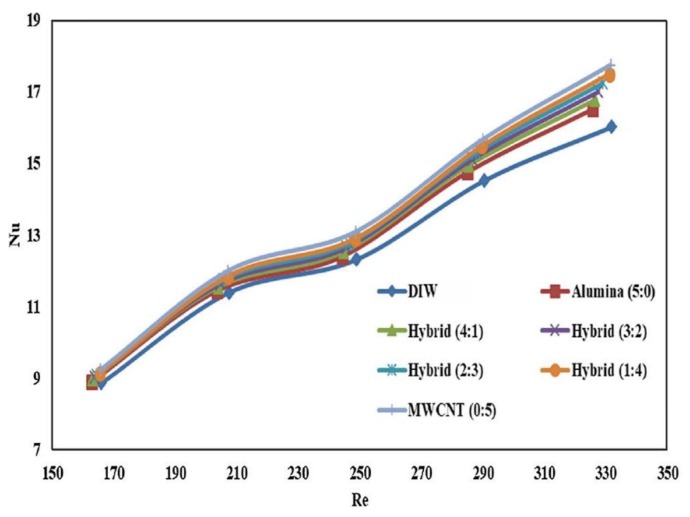
Variations of *Nu* number versus *Re* number in different ratios of the Al_2_O_3_ and MWCNT nanoparticles. Reproduced with permission from [188]. Elsevier, 2019.

**Figure 16 nanomaterials-10-00734-f016:**
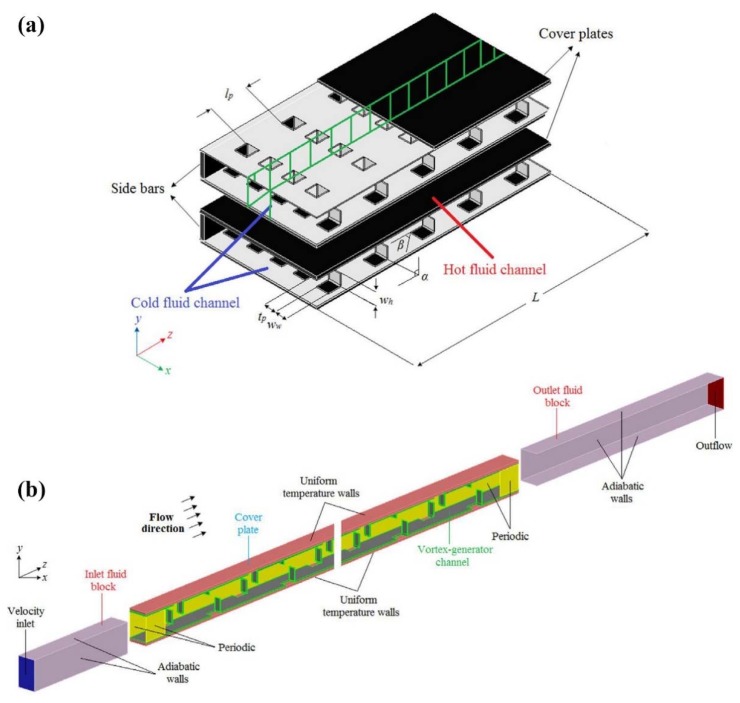
A schematic view of the: (**a**) studied plate-fin heat exchanger equipped with vortex-generator, and (**b**) the computation domain. Reproduced with permission from [204]. Elsevier, 2014.

**Figure 17 nanomaterials-10-00734-f017:**
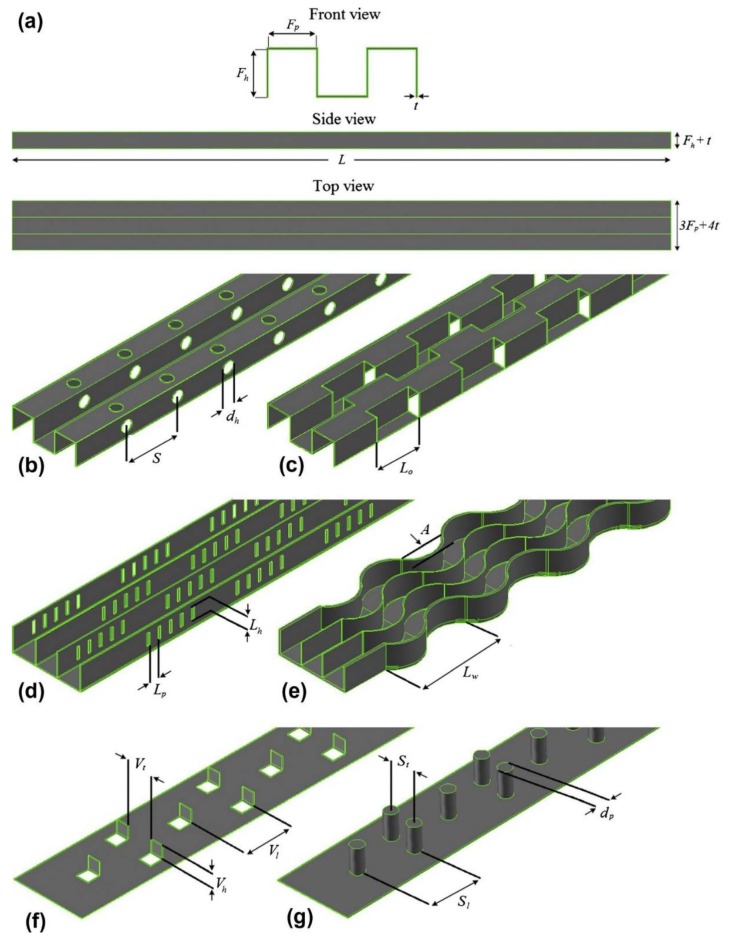
A schematic view of the different studied plate-fin channel; (**a**) plain channel, (**b**) perforated channel, (**c**) offset strip channel, (**d**) louvered channel, (**e**) wavy channel, (**f**) vortex-generator channel, and (**g**) pin channel. Reproduced with permission from [205]. Elsevier, 2014.

**Figure 18 nanomaterials-10-00734-f018:**
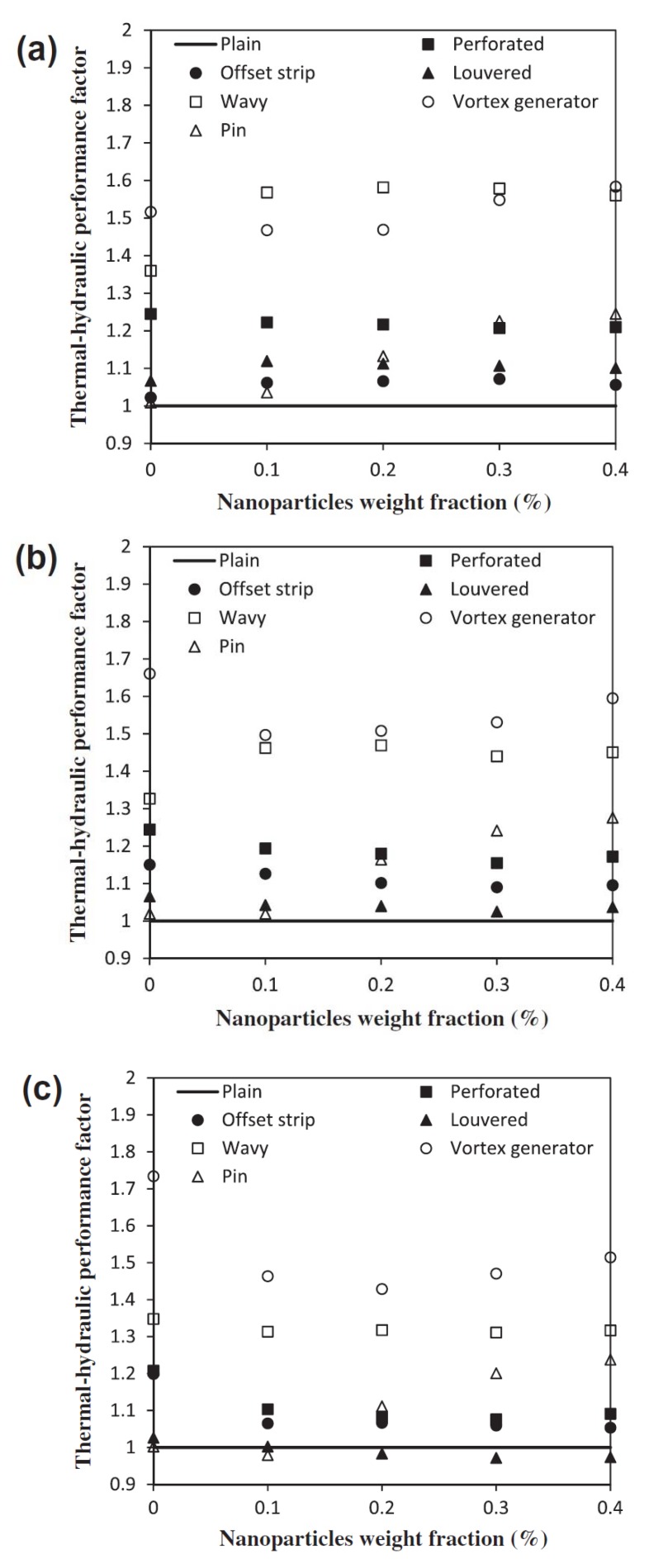
The variations of the thermal-hydraulic performance of different studied channels versus solid concentration in different flow rates: (**a**) 2 lpm, (**b**) 3.5 lpm, and (**c**) 5 lpm. Reproduced with permission from [205]. Elsevier, 2014.

**Figure 19 nanomaterials-10-00734-f019:**
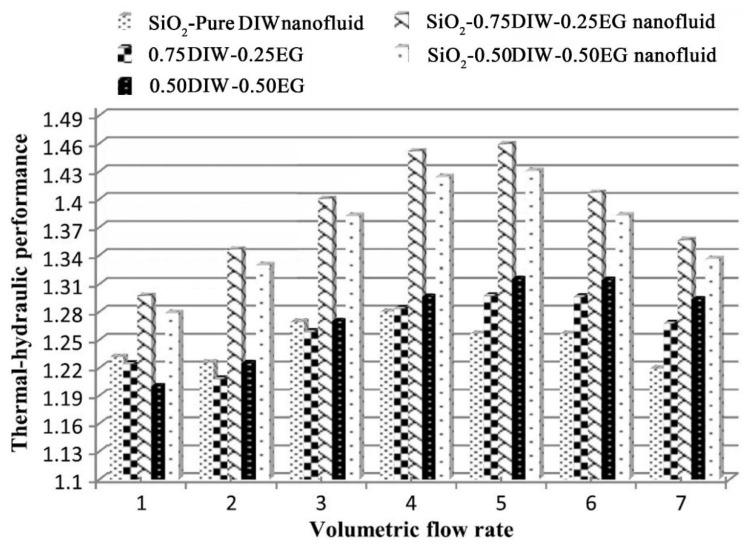
Variations of the thermal-hydraulic performance versus volumetric flow rate for different ratios of DIW-EG mixture as the basefluid for the SiO_2_-based nanofluid [206].

**Figure 20 nanomaterials-10-00734-f020:**
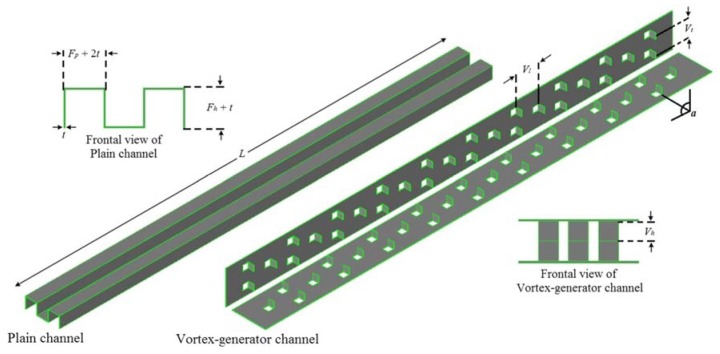
The studied geometry of the plain and vortex-generator channel. Reproduced with permission from [207]. Springer Nature, 2015.

**Figure 21 nanomaterials-10-00734-f021:**
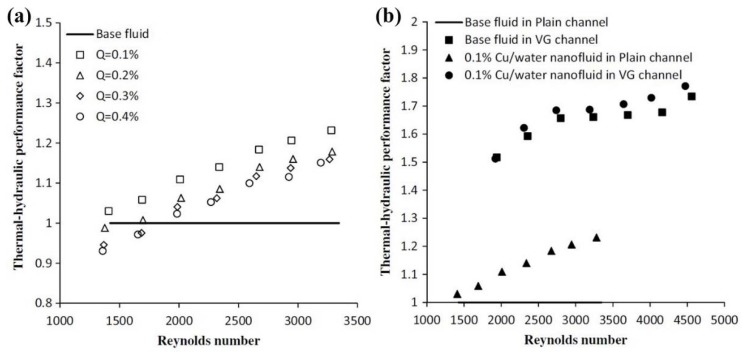
Variations of the thermal-hydraulic performance versus *Re* number for: (**a**) different nanofluids flow inside plain channel, and (**b**) different heat transfer enhancement methods. Reproduced with permission from [207]. Springer Nature, 2015.

**Table 1 nanomaterials-10-00734-t001:** Different types of nanoparticles and basefluids used in fabricating nanofluids.

Origin	Nanoparticles	Basefluids	Source
Metals	Cu *	Water, EG, oil, acetone, and water & EG mixture.	[68,74,75,76,77,78,79]
	Ag *	Water, and toluene.	[80,81]
	Au *	Water, and toluene.	[81,82,83]
	Al *	Water, oil, EG, kerosene.	[15,84,85,86,87]
Oxides	Al_2_O_3_ *	Water, EG, oil, and water & glycerine mixture.	[82,88,89,90,91,92,93,94,95,96]
	CuO *	Water, oil, and R-134a *.	[92,93,97,98,99,100,101]
	ZnO *	Water, EG, and oil.	[102,103,104,105,106,107,108,109]
	TiO_2_ *	Water, EG, oil, water & EG mixture, and bioglycol & water mixture.	[110,111,112,113,114,115,116]
	SiO_2_ *	Water, EG, glycerol, oil, and glycerol & EG mixture.	[80,117,118,119,120,121]
Carbon-based	MWCNTs *	Water, EG, water & EG mixture, and fullerenes oil.	[88,122,123,124,125,126,127,128]
	DWCNTs *	Water, and EG.	[129,130]
	SWCNTs *	Water, water & EG mixture.	[131,132]
	Nanodiamond	Water, EG, propylene glycol, midel oil, silicone oil, mineral oil, transformer oil, and engine oil.	[133]
	Graphene	Water, water & EG mixture.	[134,135,136]
	Graphite	Water, texatherm oil.	[137,138,139]

(*) Note: Cu, Ag, Au, Al, Al_2_O_3_, CuO, ZnO, TiO_2_, SiO_2_, MWCNTs, DWCNTs, SWCNTs, and R-134a are referred to copper, silver, gold, aluminium, aluminium oxide (also known as alumina), copper oxide (also known as cupric oxide), zinc oxide, titanium oxide, silicon dioxide (also known as silica), multi-walled CNTs, double-walled CNTs, single-walled CNTs, and 1,1,1,2-Tetrafluoroethane, respectively.

**Table 2 nanomaterials-10-00734-t002:** A summary of the recently published literature on the effects of using nanofluids on heat transfer performance and pressure drop of PHEs.

Reference	Nanofluid	Considered Conditions and Objectives	Type of HE	Findings
Tiwari et al. [196]	CeO_2_-water*	- Solid concentrations: 0.5 to 3 vol. % - Flow rates: 1 to 4 lpm - The effects of nanofluid on heat transfer and pressure drop.	Chevron corrugated PHE	They found that the optimum solid concentration (0.75 vol. %) in which the heat transfer reached its maximum enhancement by 39%. They reported that increasing the flow rate of the nanofluid and the hot water leads to enhancing the heat transfer coefficient. Moreover, the increase in the pressure drop at the optimum solid concentration is negligible while the heat transfer has been significantly improved.
Barzegarian et al. [190]	TiO_2_-water	- Solid concentrations: 0.3 to 1.5 wt. % - Flow regime: turbulent - Re numbers: 159 to 529 - Effects of solid concentration and Re number on heat transfer and pressure drop.	Brazed PHE	Their results revealed that increasing the *Re* number and solid concentration results in enhancing the convective heat transfer coefficient, and the maximum enhancement took place at the highest solid concentration by 23.7%. They also reported that the increase in pressure drop by increasing the solid concentration is negligible.
Kumar et al. [191]	ZnO-water	- Solid concentrations: 0.5 to 2.0 vol. % - The effects of nanofluid on heat transfer performance and finding the optimum solid concentration.	Chevron-type PHE	They reported that the solid concentration of 1.0 vol. % is the optimum solid concentration where the maximum heat transfer rate is achieved.
Unverdi and Islamoglu [197]	Al_2_O_3_-water	- Solid concentrations: 0.25 to 1 vol. % - Flow rates: 90 to 300 kg/h - Re number: 600 to 1900 - The effects of nanofluid on heat transfer and pressure drop.	Chevron-type PHE	They reported that increasing the solid concentration and flow rate results in enhancing the *Nu* number by the maximum of 42.4%. They also reported that the maximum increase in the heat transfer and pressure drop took place at the highest solid concentration and *Re* number by 6.4% and 8.4%, respectively.
Pourhoseini et al. [192]	Ag-water	- Nanofluid concentrations: 0 to 10 mg/L - Flow rate: 2 to 8 lpm - Nanofluid inlet temperature: 36, 46, and 56 °C - The effects of flow rate and solid concentration on heat transfer performance.	CR14-45 COMER PHE	They found that the effect of flow rate on heat transfer performance is more significant than the effect of solid concentration.
Wang et al. [193]	Graphene nanoplatelets-EG/water (50:50)	- Solid concentrations: 0.01 to 1.0 wt. % - Re number: 10 to 400 - The effects of using nanofluid on heat transfer and pressure drop.	Miniature PHE	They reported the maximum enhancement of 4% in heat transfer as the solid concentration increased. Moreover, they reported that the increase in *Re* number leads to enhancing the heat transfer performance in all the studied solid concentrations. The same trend as was observed for the pressure drop; increasing the solid concentration and *Re* number leads to increasing the pressure drop.
Mansoury et al. [194]	Al_2_O_3_-water	- Solid concentrations: 0.2 to 1 vol. % - Flow regime: turbulent - The effects of nanofluid on heat transfer and pressure drop in different HEs.	Different HEs; a Double-pipe, a Shell and tube, and a PHE	They reported that the maximum heat transfer of 60% is achieved in the double-pipe HE, while the minimum enhancement took place in the PHE by 11%. Moreover, the minimum increase in pressure drop has been experienced in the PHE.
Elias et al. [198]	Al_2_O_3_-water	- Solid concentrations: 0 to 0.5 vol. % - Temperatures: 25 to 55 °C - Re numbers: 180 to 350 - The effects of using nanofluid on heat transfer performance and pressure drop.	Chevron-type PHE	The results revealed the maximum enhancement of 7.8% in the heat transfer coefficient at the solid concentration of 0.5 vol. %. Moreover, increasing the solid concentration leads to increasing the pressure drop.
Tayyab at al. [199]	CuO-water	- Solid concentrations: 0.2 to 0.6 vol. % - Flow rates: 1 to 9 lpm - Different surface roughness - The effects of nanofluid on heat transfer performance in different HEs.	Different HEs: Shell and tube, concentric, spiral, and PHE	The results revealed that the heat transfer performance of the nanofluid in the PHE is better than the other studied HEs. The maximum enhancement in heat transfer for the PHE is 26% while for the other HEs, 21% is reported.
Attalla and Maghrabie [200]	Al_2_O_3_-water	- Solid concentrations: 1.2 to 2.6 vol. % - Re numbers: 500 to 5000 - The effects of using nanofluid on the Nu number, friction factor, and heat transfer enhancement.	PHE	The results revealed that the heat transfer performance and the pressure drop has been increased as the solid concentration and surface roughness increased. Moreover, it is found that the influence of the surface roughness is more noticeable than the solid concentration.
Talari et al. [195]	Al_2_O_3_-water	- Solid concentrations: 0 to 5 vol. % - Finding the optimum solid concentration for heat transfer intensification.	Corrugated PHE	They declared that since the heat transfer enhancement of the nanofluid showed a monotonic increase, it is not possible to find an optimum solid concentration.
Sözen et al. [201]	Kaolin-water	- Solid concentration: 2 wt. % - Temperatures: 40, 45, and 50 °C - The effect of using nanofluid on heat transfer performance.	Spiral PHE	It is revealed that using nanofluid instead of the based fluid leads to having 17.6% enhancement in heat transfer rate. Moreover, increasing the *Re* number leads to decreasing the effectiveness of the PHE.
Meisam et al. [202]	Al_2_O_3_-water TiO_2_-water SiO_2_-water	- Solid concentrations: 0.05, 0.1, and 0.2 wt. % - Temperatures: 30 to 50 °C - Re numbers: 35.9 to 160.6 - Flow rates: 0.4 to 2 L/m - The effects of using different nanofluids on heat transfer performance have been studied.	PHE	The results revealed that adding nanoparticles to the basefluid leads to considerable enhancement in heat transfer performance. The maximum enhancement in the heat transfer achieved by using SiO_2_-water nanofluid at the highest solid concentration and *Re* number of 37 by 2.82%, while the minimum enhancement has been experienced by using Al_2_O_3_-water nanofluid at the solid concentration of 0.1 wt. % and *Re* number 158 by 1.64%.
Soman et al. [203]	γ-Al_2_O_3_-water*	- Solid concentrations: 0.1, 0.2, and 0.3 wt. % - Mas flow rate: 0.016-0.082 kg/s - Re number: 200 to 1100	Dimpled PHE	It is revealed that increasing the mass flow rate leads to increasing the heat transfer rate in the PHE. Moreover, increasing the mass flow rate has a direct effect on the heat transfer performance. A new correlation for predicting the Nu number has also been proposed.

(*) Note: CeO_2_, and γ-Al_2_O_3_ are referred to cerium dioxide, and the alpha phase of alumina, respectively

**Table 3 nanomaterials-10-00734-t003:** A summary of the available literature on the effects of nanofluids, vortex-generators, winglets, and perforations on the thermal-hydraulic performance of plate-fin HEs.

Reference	Nanofluid	Considered Conditions and Objectives	Type of HE	Findings
Aliabadi et al. [208]	Al_2_O_3_-water	- Considering the effects of perforations, winglets, and nanofluids on heat transfer performance. - Solid concentration: 0.1 and 0.3 wt. % - Waviness aspect ratio: 0.33 to 0.51 - Perforation diameter: 5 mm - Winglets heights and width: 5 mm - Re number: 3900 to 11,400	Wavy plate-fin HE	It is revealed that the *Nu* number for the wavy channel is higher than that of the plain channel. Moreover, the performance factor values show that applying these three techniques leads to improving the thermal-hydraulic performance of the HE.
Aliabadi et al. [209]	Al_2_O_3_-water	- Solid concentrations: 0.1–0.4 wt. % - Re number: 4500–11,500 - Waviness aspect ratio: 0.33 to 0.51 - Winglets height: 2 to 6 mm	Wavy plate-fin HE	It is reported that using the nanofluid instead of the basefluid leads to increasing the thermal performance and pressure drop by 11.3% and 6.2%, respectively. Moreover, increasing the waviness aspect ratio and winglets height results in increasing the heat transfer and pressure drop.
Aliabadi and Salami [210]	Al_2_O_3_-water	- Solid concentration: 0 to 4 wt. % - Re number: 6000 to 22,000 - The effects of nanofluids, channel height, channel length, stip length, strip pitch, and strip thickness.	Offset-strip	It is reported that the most effective factor on the thermal-hydraulic performance is the channel height. Moreover, using nanofluid results in having better thermal performance compared to the basefluid.
Aliabadi and Mortazavi [211]	Al_2_O_3_-water	- Solid concentration: 0.1 to 0.4 wt. % - Re number: 4000 to 10,000 - The effects of nanofluid, waviness aspect ratio and the arrangement of the winglets.	Chevron plate-fin HE combined with holes and winglets	It is found that the HE equipped with holes and winglets showed enhanced *Nu* number by a maximum of 1.6%. Moreover, it is reported that employing the nanofluid as the working fluid also leads to enhancing the *Nu* number. The optimum solid concentration has been reported as 0.3%.
Aliabadi et al. [212]	Al_2_O_3_-water	- Solid concentrations:0.1 and 0.3 wt. % - Re number: 100–900	Plate and plate pin fin HE	It is reported that the plate-pin fin showed better heat transfer performance and lower pressure drop. Moreover, using nanofluid leads to enhancing the heat transfer coefficient and the best performance achieved at the solid concentration of 0.3 wt. %.

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
