# Peer review of "On the Role of Nanofluids in Thermal-hydraulic Performance of Heat Exchangers—A Review"

_nanomaterials, 2020, doi:10.3390/nano10040734_

Round 1

Reviewer 1 Report

The authors describe precisely thermal-hydraulic performance of heat exchengers using nanofluds. 

It is relevant and interesting to summarize orderly results published pertinent to the state-of-the-art of heat exchanger technology and the progress towards using nanofluids for enhancing their thermal-hydraulic performance.

Original content is a systematic review. This is because there are many research papers pertinent to heat exchanger technology and the progress towards using nanofluids, but there is no systematic review.

The paper is well written. The text is clear and easy to read.

Author Response

The authors would like to thank the respected reviewer for his comments and remarks. Kindly find attached the word document containing the authors reply.

Thank you very much. 

Reviewer 2 Report

The paper can be considered for publication after the following minor revisions:

1- The authors should revise the manuscript in terms of language usage.

2- Please add a nomenclature

3- Citing the following references can enrich your introduction:

https://www.sciencedirect.com/science/article/pii/S0960148119312686

https://www.sciencedirect.com/science/article/pii/S0960148119312297

https://www.sciencedirect.com/science/article/pii/S0735193319302337

https://link.springer.com/article/10.1007/s10973-019-08757-w

https://www.emerald.com/insight/content/doi/10.1108/HFF-03-2019-0273/full/html

https://link.springer.com/article/10.1007/s10973-018-7945-9

Author Response

(The authors gave the same response as above.)

Reviewer 3 Report

  1. The novelty of this review paper should be clearly stated at the end of the introduction section.
  2. More relevant and recent literature could be cited as listed here: 
  • Applied Thermal Engineering, Vol. 85, pp.81-91, 2019.
  • International Journal of Heat and Mass Transfer, Vol. 106, pp.573-592, 2017.
  • Chemical Engineering and Processing: Process Intensification, Vol. 87, pp. 88-103, 2015.
  • International Communications in Heat and Mass Transfer, Vol.54, pp. 132-140, 2014.
  • International Communications in Heat and Mass Transfer, Vol.40, Issue 1, pp.36-46, 2013.

    3. The latest development in nanofluids thermophysical equations should be written.

    4. The future direction of this review paper could be elaborated more considering the recent development of the heat exchanger design and its applications.

Author Response

(The authors gave the same response as above.)

Round 2

Reviewer 3 Report

The paper can now be accepted as the authors have made the required amendments.